# Accounting for multiple imputation-induced variability for differential analysis in mass spectrometry-based label-free quantitative proteomics

**Marie Chion**[1,2,3]*, **Christine Carapito**[2,4], **Frédéric Bertrand**[1,5]

**1** Institut de Recherche Mathématique Avancée, UMR 7501, CNRS-Université de Strasbourg, Strasbourg, France, **2** Laboratoire de Spectrométrie de Masse Bio-Organique, Institut Pluridisciplinaire Hubert Curien, UMR 7178, CNRS-Université de Strasbourg, Strasbourg, France, **3** Laboratoire Mathématiques appliquées à Paris 5, UMR 8145, CNRS-Université Paris Cité, Paris, France, **4** Infrastructure Nationale de Protéomique ProFi - FR2048, 67087 Strasbourg, France, **5** Laboratoire Informatique et Société Numérique, Université de Technologie de Troyes, Troyes, France

* marie.chion@protonmail.com

**Data Availability Statement:** Our mi4p algorithm is implemented under the R environment in the mi4p package that is publicly available on the CRAN. The development version, as well as the R

## Abstract

Imputing missing values is common practice in label-free quantitative proteomics. Imputation aims at replacing a missing value with a user-defined one. However, the imputation itself may not be optimally considered downstream of the imputation process, as imputed datasets are often considered as if they had always been complete. Hence, the uncertainty due to the imputation is not adequately taken into account. We provide a rigorous multiple imputation strategy, leading to a less biased estimation of the parameters' variability thanks to Rubin's rules. The imputation-based peptide's intensities' variance estimator is then moderated using Bayesian hierarchical models. This estimator is finally included in moderated *t*-test statistics to provide differential analyses results. This workflow can be used both at peptide and protein-level in quantification datasets. Indeed, an aggregation step is included for protein-level results based on peptide-level quantification data. Our methodology, named `mi4p`, was compared to the state-of-the-art `limma` workflow implemented in the `DAPAR` R package, both on simulated and real datasets. We observed a trade-off between sensitivity and specificity, while the overall performance of `mi4p` outperforms `DAPAR` in terms of *F*-Score.

## Author summary

Statistical inference methods commonly used in quantitative proteomics are based on the measurement of peptide intensities. They allow the deduction of protein abundances provided that sufficient peptides per protein are available. However, they do not satisfactorily consider peptides or proteins whose intensities are missing under certain conditions, even though they are particularly interesting from a biological or medical point of view, since

scripts which led to the results presented, can also be found on a GitHub repository (https://github.com/mariechion/mi4p). The spiked yeast dataset and the Arabidopsis thaliana spiked dataset are public and accessible on the ProteomeXchange website using the identifiers PXD003841 and PXD027800.

**Funding:** This work was funded through a PhD grant (2018-2021) awarded to MC and received by FB and CC from the Agence Nationale de la Recherche (ANR) through the Labex IRMIA [ANR-11-LABX-0055 IRMIA]. The funders had no role in study design, data collection and analysis, decision to publish, or preparation of the manuscript.

**Competing interests:** The authors have declared that no competing interests exist.

they may explain a difference between the groups being compared. Some state-of-the-art statistical proteomics data processing software proposes to impute these missing values, while others simply remove proteins with too many missing peptides. The statistical treatment is not entirely satisfactory when imputation methods are used, notably multiple imputation techniques. Indeed, even if these statistical tools are relevant in this context, the data sets once imputed are considered as having always been complete in the subsequent analyses: the uncertainty caused by the imputation is not taken into account. These analyses generally conclude with a study of the differences in protein abundances between the different conditions, either using Student's or Welch's test for the most rudimentary approaches or using the $t$-tempered testing techniques based on empirical Bayesian approaches. Thus, we propose a new methodology that starts by imputing missing values at the peptide level and estimating the uncertainty associated with this imputation and naturally extends by incorporating this uncertainty into the current moderated variance estimation techniques.

This is a *PLOS Computational Biology* Methods paper.

## Introduction

Dealing with incomplete data is one of the main challenges as far as statistical analysis is concerned. Different strategies can be used to tackle this issue. The simplest way consists of deleting from the dataset the observations for which there are too many missing values, leading to a complete-case dataset. However, it causes information loss, might create bias, and ultimately could result in poorly informative datasets. Some methods combining qualitative and quantitative statistical tools can also be considered [1]. Another way to cope with missing data is to use methods that account for the missing information. For the last decades, researchers advocated the use of a single technique called imputation. Imputing missing values consists of replacing a missing value with a value derived using a user-defined formula (such as the mean, the median or a value provided by an expert, thus considering the user's knowledge). Hence it makes it possible to perform the analysis as if the data were complete. More particularly, the vector of parameters of interest can be then estimated. Single imputation means completing the dataset once and considering the imputed dataset as if it was never incomplete, see Fig 1. However, single imputation has the major disadvantage of discarding the variability from the missing data and the imputation process. It may also lead to a biased estimator of the vector of parameters of interest.

Multiple imputation [2] closes this loophole by generating several imputed datasets. These datasets are then used to build a combined estimator of the vector of parameters of interest by usually using the mean of the estimates among all the imputed datasets, see Fig 2. This combined estimator is known as the first Rubin's rule. The second Rubin's rule states a formula to estimate the variance of the combined estimator, decomposing it as the sum of the intra-imputation variance component and the between-imputation component. The rule of thumb takes the number of imputed datasets as the percentage of missing values in the original dataset [3]. Recent work focused on better estimating the Fraction of Missing Information [4] or improving that rule [5]. Note that Rubin's rules cannot be used in order to get a combined imputed

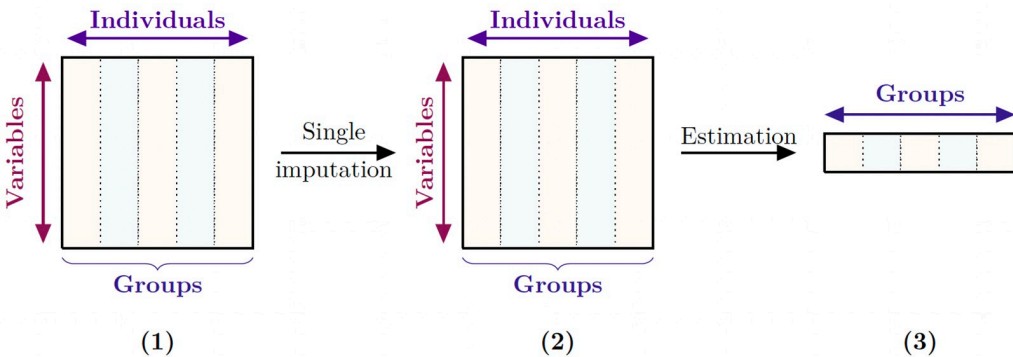

**Fig 1. Single imputation strategy. (1)** Initial dataset with missing values. It is supposed to be made of N observations that are split into K groups. **(2)** Single imputation provides an imputed dataset. **(3)** The vector of parameters of interest is estimated based on the single imputed dataset.

dataset but instead provide an estimator of the vector of parameters of interest and an estimator of its covariance matrix, both based on the multiple imputation process, see Fig 2.

Dealing with missing values is also one of the main struggles in label-free quantitative proteomics. Intensities of thousands of peptides are obtained by liquid chromatography-tandem mass spectrometry, using extracted ion chromatograms. Missing peptides' intensities arise from various reasons (biological, analytical, bioinformatical) and obey different missing values mechanisms. For example, the considered peptide is missing in the given biological sample, and the intensity is then missing not at random (MNAR) or it could have not been accurately identified (non searched biochemical modification or peptides co-elution, . . .) and the intensity is then missing at random (MAR).

In state-of-the-art software for statistical analysis in label-free quantitative proteomics, single imputation is the most commonly used method to deal with missing values. The `MSstats` `R` package (available on Bioconductor) [6] distinguishes missing completely at random values from missing values due to low intensities. The user can then choose to impute the censored value using either a threshold value or an Accelerated Failure Time model. The Perseus software [7] offers three methods for single imputation: either imputing by "NaN"(hence ignoring missing values in downstream analysis), impute by a user-defined constant or impute according to a Gaussian distribution in order to simulate intensities, which are lower than the limit of

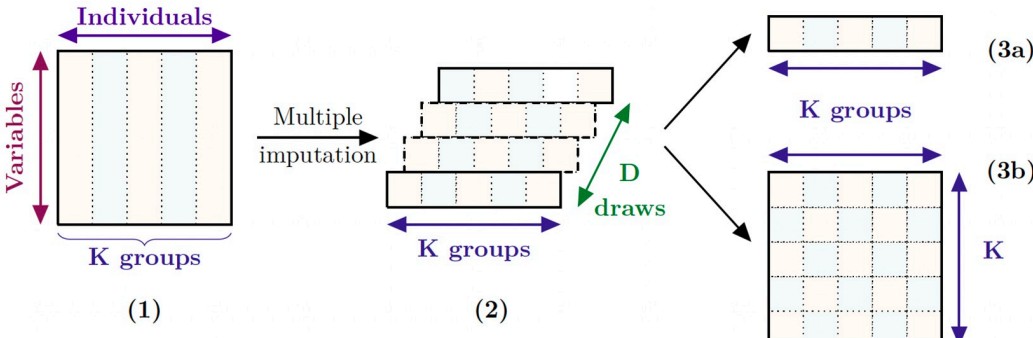

**Fig 2. Multiple imputation strategy. (1)** Initial dataset with missing values. It is supposed to have N observations that are split into K groups. **(2)** Multiple imputation provides D estimators for the vector of parameters of interest. **(3a)** The D estimators are combined using the first Rubin's rule to get the combined estimator. **(3b)** The estimator of the variance-covariance matrix of the combined estimator is provided by the second Rubin's rule.

detection. As far as machine learning is concerned, a method for imputing missing values in label-free mass spectrometry-based proteomics datasets was suggested [8]. Note that the authors of the `MSqRob R` package recently proposed to bypass imputation with a hurdle model that combines count-based differential detection with intensity-based differential abundance [9].

The ProStaR software based on the `DAPAR R` package splits missing values into two categories, whether they are Missing in an Entire Condition (MEC) or Partially Observed Values (POV) and allow them to be imputed using different methods [10, 11]. The software allows single imputation, using either a small quantile from the distribution of the considered biological sample, the *k*-Nearest Neighbours (kNN) algorithm or the Structured Least Squares Adaptative algorithm or by choosing a fixed value. The PANDA-view software [12] also enables the use of the kNN algorithm or a fixed value. Moreover, both software programs allow the possibility of imputing the dataset several times before combining the imputed datasets to get a final dataset without any missing values. PANDA-view relies on the `mice R` package [13], whereas ProStaR accounts for the nature of missing values and imputes them with the `imp4p R` package [14, 15]. However, both software programs consider the final dataset as if it had always been complete. The uncertainty due to multiple imputation is not properly taken into account downstream of the imputation step.

In the following, we will conduct the multiple imputation process to its end and use the imputed datasets to provide a combined estimator of the vector of parameters of interest as well as a combined estimator of its variance-covariance matrix estimator. We will then project this matrix to get a unidimensional variance estimator before moderating it using the empirical Bayes procedure [16, 17]. It is well known that such a moderating step highly improves the following statistical analyses such as significance testing of confidence interval estimation, both at the peptide level [18, 19] or the protein level [19, 20].

## Methods

### Multiple imputation algorithms

Several methods for imputing missing values in mass spectrometry-based proteomics datasets were developed in the last decade. However, the recent benchmarks of imputation algorithms do not reach a consensus (as shown in S1 Table). This is mainly due to the complex nature of the underlying missing values mechanism. This work focuses on some of the most commonly used methods, which are described in Table 1. The *k*-nearest neighbours (kNN) method [21–23] imputes missing values by averaging the *k*-nearest observations of the given missing value in terms of Euclidean distance. The Maximum Likelihood Estimation (MLE) method imputed missing values using the EM algorithm [15, 24]. The Bayesian linear regression (norm) method imputes missing values using the normal model and following the method described and implemented in the `mice R` package [25, 26]. Some methods implemented in the `imp4p R` package [15] were also considered, namely principal component analysis (PCA) [27] and

**Table 1. Overview of the imputation methods considered in this work.**

| Method | Implementation | References |
|---|---|---|
| *k*-nearest neighbours | `impute.knn` (impute R package) | [21–23] |
| Maximum likelihood estimation | `impute.mle` (imp4p R package) | [15, 24, 26] |
| Bayesian linear regression | `mice` (mice R package) | [24, 25] |
| Principal component analysis | `impute.pca` (imp4p R package) | [15, 27] |
| Random forests | `impute.RF` (imp4p R package) | [15, 28] |

random forests (`RF`) method [28]. We repeated the imputation process $D$ times to obtain $D$ imputed datasets for each method considered. We set the number of draws $D$ equal to the ceiling of the proportion of missing values in the dataset [3]. Note that if the proportion of missing values is less than 1%, the number of draws is set to 2.

## Estimation of the parameters of interest

The objective of multiple imputation is to estimate from $D$ drawn datasets the vector of parameters of interest $\boldsymbol{\beta_p} = (\beta_{p1}, \ldots, \beta_{pK})$ (*e.g.* being the vector of coefficients of the linear model for peptide $p$) and its variance-covariance matrix $\Sigma_{\boldsymbol{p}}$. Notably, accounting for multiple-imputation-based variability is possible thanks to Rubin's rules, which provide an accurate estimation of these parameters [25]. Hence, the first Rubin's rule provides the combined estimator of $\boldsymbol{\beta_p}$:

$$\hat{\boldsymbol{\beta}}_{\boldsymbol{p}} = \frac{1}{D}\sum_{d=1}^{D}\hat{\boldsymbol{\beta}}_{\boldsymbol{p,d}}, \tag{1}$$

where $\hat{\boldsymbol{\beta}}_{\boldsymbol{p,d}}$ is the estimator of $\boldsymbol{\beta_p}$ in the $d$-imputed dataset. The second Rubin's rule gives the combined estimator of the variance-covariance matrix for each estimated vector of parameters of interest for peptide $p$ through the $D$ imputed datasets such as:

$$\hat{\boldsymbol{\Sigma}}_p = \frac{1}{D}\sum_{d=1}^{D}W_d + \frac{D+1}{D(D-1)}\sum_{d=1}^{D}(\hat{\boldsymbol{\beta}}_{\boldsymbol{p,d}} - \hat{\boldsymbol{\beta}}_{\boldsymbol{p}})^T(\hat{\boldsymbol{\beta}}_{\boldsymbol{p,d}} - \hat{\boldsymbol{\beta}}_{\boldsymbol{p}}), \tag{2}$$

where $W_d$ denotes the variance-covariance matrix of $\hat{\boldsymbol{\beta}}_{\boldsymbol{p,d}}$, *i.e.* the variability of the vector of parameters of interest as estimated in the $d$-th imputed dataset.

## Projection of the covariance matrix

State-of-the-art tests, including Student's $t$-test, Welch's $t$-test and moderated $t$-test, rely on the variance estimation. Here, the variability induced by multiple imputation is described by a variance-covariance matrix, given by Eq 2. Therefore, a projection step is required to get a univariate variance parameter. Rubin's second rule decomposes the variability of the combined dataset as the sum of the within-imputation variability and the between-imputation variability. Thus, analytes whose values have been imputed should have a greater variance estimation than if the multiple imputation-induced variability was not accounted for. This amounts to "penalising" analytes for which intensity values were not observed and subsequently imputed. Hence, the projection method needs to be wisely chosen. In our work, we chose to perform projection using the following formula:

$$\hat{s}_p = \max_{k}(\hat{\Sigma}_{p,(k,k)}\mathbf{X}^t\mathbf{X}) \tag{3}$$

where $\hat{\Sigma}_{p,(k,k)}$ is the $k$-th diagonal element of the matrix $\hat{\Sigma}_p$ and $\mathbf{X}$ is the design matrix. Nevertheless, it is to be noted that this choice for the projection method is not without consequences. Indeed, this method greatly penalises imputed analytes. However, analytes that show high variance estimations might be wrongly considered non differentially expressed, as their distributions in each condition to be compared can overlap.

## Hypotheses testing

In our work, we focus our methodology on the moderated $t$-test [16] that relies on the empirical Bayes procedure, commonly used in microarray data analysis, and to a more recent extent

for differential analysis in quantitative proteomics [10]. Hence, we consider the following Bayesian hierarchical model:

$$
\begin{cases}
\hat{\sigma}_p^2 \mid \sigma_p^2 \sim \dfrac{\sigma_p^2}{d_p} \times \chi_{d_p}^2 \\[2mm]
\dfrac{1}{\sigma_p^2} \sim \dfrac{1}{d_0 \times s_0^2} \times \chi_{d_0}^2
\end{cases}
\tag{4}
$$

where $\sigma_p^2$ is the peptide-wise variance, $d_p$ is the residual degrees of freedom for the linear model for peptide $p$, $d_0$ and $s_0$ are hyperparameters to be estimated [17]. This leads to the following posterior distribution of $\dfrac{1}{\sigma_p^2}$ conditional to $\hat{\sigma}_p^2$:

$$
\frac{1}{\sigma_p^2} \mid \hat{\sigma}_p^2 \sim \frac{1}{d_p \times \hat{\sigma}_p^2 + d_0 \times s_0^2} \chi_{d_p + d_0}^2
\tag{5}
$$

From there, a so-called moderated variance estimator $\hat{\sigma}_{p[\mathrm{mod}]}^2$ of the variance $\sigma_p^2$ is derived from the posterior mean:

$$
\hat{\sigma}_{p[\mathrm{mod}]}^2 = \frac{d_p \times \hat{\sigma}_p^2 + d_0 \times s_0^2}{d_p + d_0}
\tag{6}
$$

This estimator $\hat{\sigma}_{p[\mathrm{mod}]}^2$ is then computed in the test statistic associated to the null hypothesis $\mathcal{H}_0 : \beta_{pj} = 0$, by replacing the usual sample variance by $\hat{\sigma}_{p[\mathrm{mod}]}^2$ into to the classical $t$-statistic (see Eq 7). Therefore, the results of this testing procedure account both for the specific structure of the data and the uncertainty caused by the multiple imputation step.

$$
T_{pj[\mathrm{mod}]} = \frac{\hat{\beta}_{pj}}{\sqrt{\hat{\sigma}_{p[\mathrm{mod}]}^2 (\boldsymbol{X}^{\mathrm{T}}\boldsymbol{X})_{j,j}^{-1}}}
\tag{7}
$$

with $(\boldsymbol{X}^{\mathrm{T}}\boldsymbol{X})_{j,j}^{-1}$ the $j$-th diagonal element in the matrix $(\boldsymbol{X}^T\boldsymbol{X})^{-1}$ and $\hat{\beta}_{pj}$ is the $j$-th coefficient of the linear model for peptide $p$. Under the null hypothesis $\mathcal{H}_0$, $T_{pj[\mathrm{mod}]}$ is assumed to follow a Student distribution with $d_p + d_0$ degrees of freedom.

As there are as many tests performed as the number of peptides considered, the proportion of falsely rejected hypotheses has to be controlled. Here, the Benjamini-Hochberg False Discovery Rate control procedure was performed using the cp4p R package [29, 30].

Note that the implementation of the aforementioned testing framework strongly relies on the limma R package. Hence, this work can be generalised to any experimental design.

## Aggregation

The methodology implemented in the mi4p R package can be applied to peptide-level quantification data as well as protein-level quantification data. However, common practice in proteomics consists in inferring results at the protein level from peptide-level data. In particular, imputation should be performed at the peptide level, before aggregating peptides into proteins [31]. Therefore, we adjusted our pipeline as follows:

1. Out-filtration of non-unique peptides from the peptide-level quantification dataset.

2. Normalisation of the $\log_2$-transformed peptide intensities.

3. Multiple imputation of $\log_2$-transformed peptide intensities.

4. Aggregation by summing all peptides intensities (non-$\log_2$-transformed) from a given protein in each imputed dataset.

5. $\log_2$-transformation of protein intensities.

6. Estimation of variance-covariance matrix.

7. Projection of the estimated variance-covariance matrix.

8. Moderated $t$-testing on the combined protein-level dataset

## Indicators of performance

We compared our methodology to the `limma` testing pipeline implemented in the state-of-the-art `ProStaR` software, through the `DAPAR` R package, as described in Fig 3. To assess the performances of both methods, we used the following measures: sensitivity (also known as true positive rate or recall), specificity (also known as true negative rate), precision (also known as positive predictive value), $F$-score and Matthews correlation coefficient. In our work, we define a true positive (respectively negative) as a peptide/protein that is correctly considered as (not) differentially expressed by the testing procedure. Similarly, we define a false positive (respectively negative) as a peptide/protein that is falsely considered as (not) differentially expressed by the testing procedure.

## Results and discussion

### Simulated datasets under missing at random assumption

**Simulation designs.**  We evaluated our methodology on three types of simulated datasets. First, we considered an experimental design where the distributions of the two groups to be compared scarcely overlap. This design led to a fixed effect one-way analysis of variance model (ANOVA), which can be written as:

$$y_{pnk} = \mu + \delta_{pk} + \epsilon_{pnk} \tag{8}$$

with $\mu = 100$, $\delta_{pk} = 100$ if $1 \leq p \leq 10$ and $k = 2$ and $\delta_{pk} = 0$ otherwise and $\epsilon_{pnk} \sim \mathcal{N}(0, 1)$. Here, $y_{pnk}$ represented the log-transformed abundance of peptide $p$ in the $n$-th sample. Thus, we generated 100 datasets by considering 200 individuals and 10 variables, divided into 2 groups of 5 variables, using the following steps:

1. For the first 10 rows of the data frame, set as differentially expressed, draw the first 5 observations (first group) from a Gaussian distribution with a mean of 100 and a standard deviation of 1. Then draw the remaining 5 observations (second group) from a Gaussian distribution with a mean of 200 and a standard deviation of 1.

2. For the remaining 190 rows, set as non-differentially expressed, draw the first 5 observations as well as the last 5 observations from a Gaussian distribution with a mean of 100 and a standard deviation of 1.

Secondly, we considered an experimental design, where the distributions of the two groups to be compared might highly overlap. Hence, we based it on the random hierarchical ANOVA model [31, 32]. The simulation design followed the following model:

$$y_{pnk} = P_p + G_{pk} + \epsilon_{pnk} \tag{9}$$

where $y_{pnk}$ is the log-transformed abundance of peptide $p$ in the $n$-th sample, $P_p$ is the mean

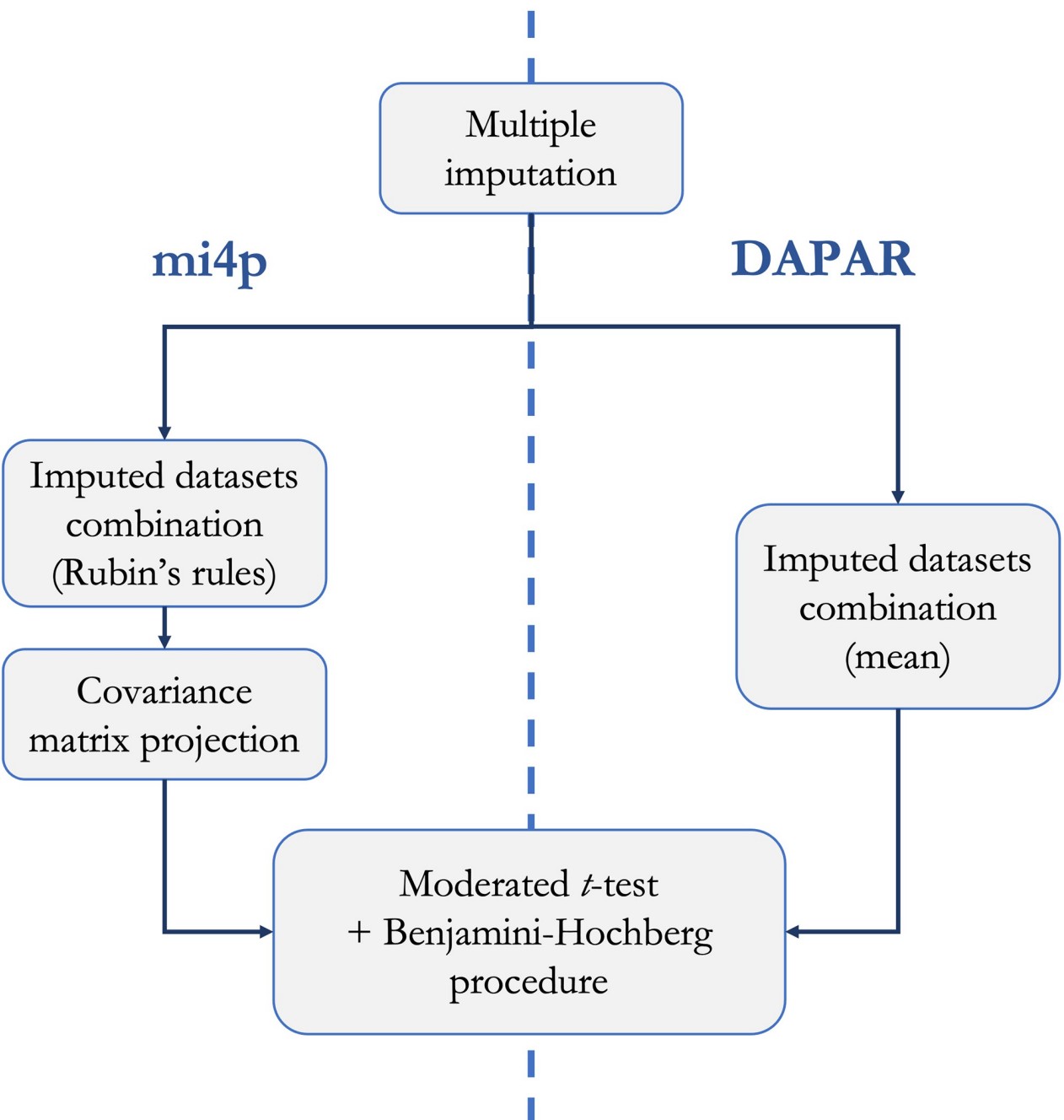

**Fig 3. Workflow conducted for performance evaluation of the `mi4p` methodology and comparison to the one implemented in the `DAPAR` R package.**

value of peptide $p$, $G_{pk}$ is the mean differences between the condition groups, and $\epsilon_{pnk}$ is the random error terms, which stands for the peptide-wise variance. We generated 100 datasets by considering 1000 individuals and 20 variables, divided into 2 groups of 10 variables, using the following steps:

1. Generate the peptide-wise effect $P_p$ by drawing 1000 observations from a Gaussian distribution with a mean of 1.5 and a standard deviation of 0.5.

2. Generate the group effect $G_{pk}$ by drawing 200 observations (for the 200 individuals set as differentially expressed) from a Gaussian distribution with a mean of 1.5 and a standard deviation of 0.5 and 800 observations fixed to 0.

3. Build the first group dataset by replicating 10 times the sum of $P_p$ and the random error term, drawn from a Gaussian distribution of mean 0 and standard deviation 0.5.

4. Build the second group dataset by replicating 10 times the sum of $P_p$, $G_{pk}$ and the random error term drawn from a Gaussian distribution of mean 0 and standard deviation 0.5.

5. Bind both datasets to get the complete dataset.

Finally, we considered an experimental design similar to the second one, but with random effects $P_p$ and $G_{pk}$. The 100 datasets were generated as follows.

1. For the first group, replicate 10 times (for the 10 variables in this group) a draw from a mixture of 2 Gaussian distributions. The first one has the following parameters: a mean of 1.5 and a standard deviation of 0.5 (corresponds to $P_p$). The second one has the following parameters: a mean of 0 and a standard deviation of 0.5 (corresponds to $\epsilon_{pnk}$).

2. For the second group replicate 10 times (for the 10 variables in this group) a draw from a mixture of the following 3 distributions.

    1. The first one is a Gaussian distribution with the following parameters: a mean of 1.5 and a standard deviation of 0.5 (corresponds to $P_p$).

    2. The second one is the mixture of a Gaussian distribution with a mean of 1.5 and a standard deviation of 0.5 for the 200 first rows (set as differentially expressed) and a zero vector for the remaining 800 rows (set as not differentially expressed). This mixture illustrates the $G_{pk}$ term in the previous model.

    3. The third distribution has the following parameters: a mean of 0 and a standard deviation of 0.5 (corresponds to $\epsilon_{pnk}$).

All simulated datasets were then amputed to produce MAR missing values in the following proportions: 1%, 5%, 10%, 15%, 20% and 25%.

**Comparison of imputation methodologies.**   To compare the imputation methods considered in Table 1, we used the synthetic data from the aforementioned second set of MAR simulations. Let us highlight that reviews on imputation methods evaluation often base their study on real datasets by subsetting them to complete data and amputating them afterwards (S1 Table). However, such approaches remain limited, as the parameters of the data cannot be controlled. Recall that we simulated 100 datasets, which were amputated afterwards. Hence both imputed and real values can be accessed. In this section, we aim at evaluating the potential bias that can arise from the imputation process. We based our comparison on the amputated datasets with a proportion of missing values of 10%, so we impute each dataset $D_Q = 10$ times. Consider then the set of all missing values coming from the $Q = 100$ datasets. Let $n_q$ denote the number of missing values in the $q$-th dataset, with $1 \leq q \leq Q$. The set of all missing data is then constituted of $N_Q = \sum_{q=1}^{Q} n_q$ elements. In our work, we take the number of draws for multiple imputation as the percentage of missing values. Therefore, multiple imputation produces ten vectors of size $N_Q$ corresponding to the ten draws of the considered vector.

**Imputation error for each draw.**   To evaluate the performance of the imputation methodologies considered, we first consider the error on each draw. Let $y_i$ denote the $i$-th value in the

previously defined set and $y_i^{(d)}$ the $d$-th draw for $y_i$. Hence, we define the error $\varepsilon_i^{(d)}$ for each imputed value $y_i^{(d)}$ as:

$$\varepsilon_i^{(d)} = y_i^{(d)} - y_i, \forall i \in 1, \cdots, N_Q, \forall d \in 1, \cdots, D_Q.$$

The $D_Q \times N_Q$ errors are calculated for all imputation method considered, namely kNN, MLE, norm, PCA and RF (detailed in Table 1). To compare the performances of these methods, Fig 4 summarises the distributions of $(\varepsilon_i^{(d)})_{i=1,\ldots,N_q,d=1,\ldots,D_Q}$ for the five imputation methods considered. First, it is comforting to observe that the errors are all centred on zero. Moreover, let us also point out that the MLE and norm methods provide a slightly increased variability compared to other methods. The kNN, PCA and RF methods show equivalent performance as far as single imputation is concerned.

**Imputation error for the mean of draws.** Following the first Rubin's rule (Eq 1), the $D_Q$ drawn datasets are combined using the mean. In order to provide additional insights about the empirical errors of the different multiple-imputation procedures, let us compute the differences between the averaged imputed values used in practice and the actual values. For each imputation method, the errors are averaged over the $D_Q$ draws (corresponding to the $D_Q$

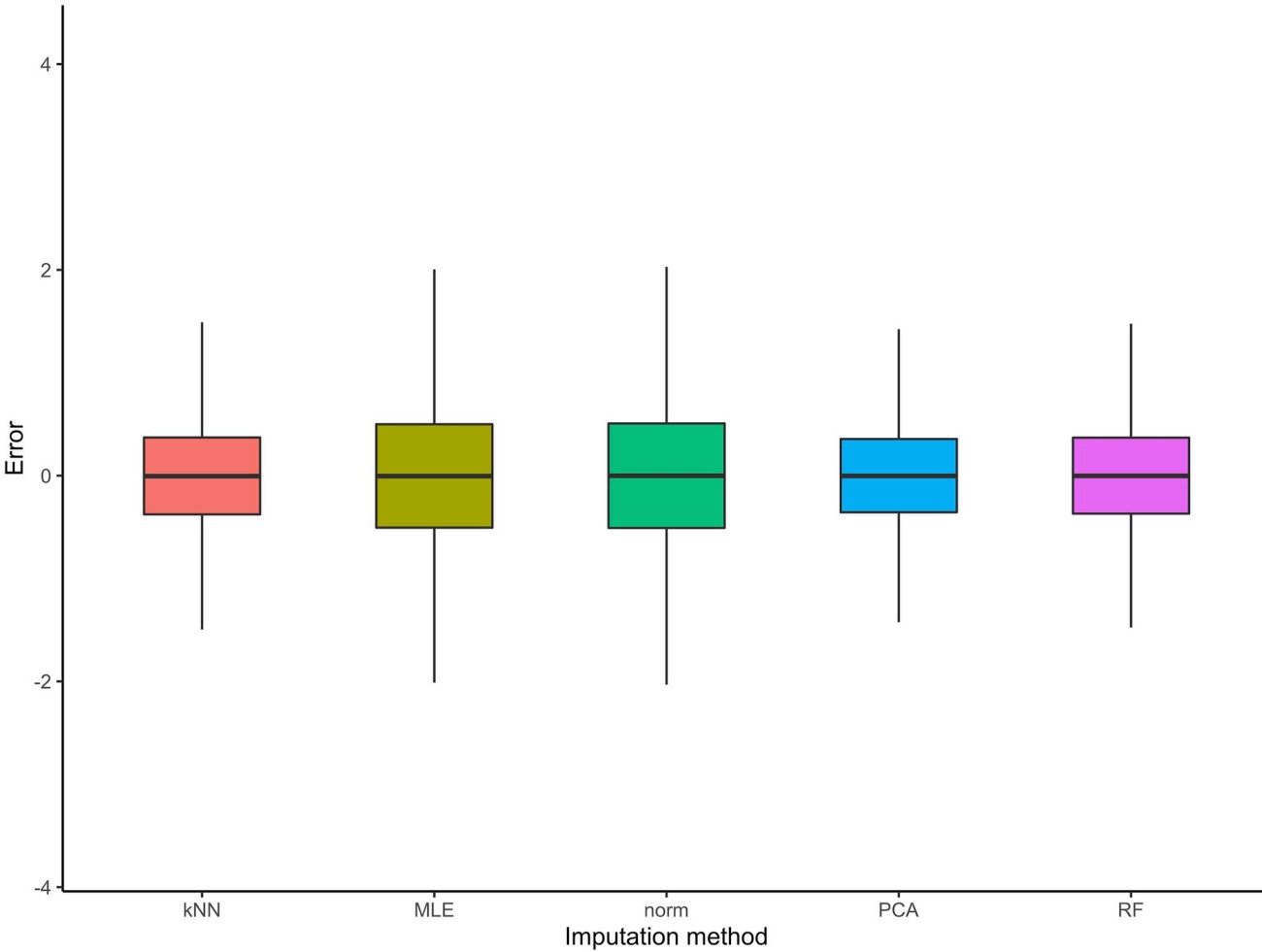

**Fig 4. Distribution of empirical errors for the five imputation methods considered on the second set of MAR simulations.**

different imputations), which we expect to stabilise the error values. In contrast to the previous approach, the associated formula becomes:

$$\varepsilon_i = \frac{1}{D_Q}\sum_{d=1}^{D_Q} y_i^{(d)} - y_i, \ \ \forall i \in 1, \cdots, N_Q.$$

Fig 5 suggests equivalent performance for all five methods as far as the mean of all imputed datasets is concerned. In terms of variability, we can still observe a slightly increased interquartile range for the MLE imputation method.

**Computation time.**   As a complement to determine the advantages of each approach, we compared the running time of all imputation processes. Therefore, we considered the total time needed for imputing each simulated dataset $D_Q$ times. The boxplots on Fig 6 highlight the MLE and kNN method to be the fastest.Compared to MLE imputation method, the PCA method is on average 3.5 times slower and the norm and RF methods are respectively on average 7.4 times and 8.1 times slower. At this stage of the comparison, as all imputation methods exhibit comparable performances in terms of imputation bias, a preference can be drawn for the kNN and MLE methods.

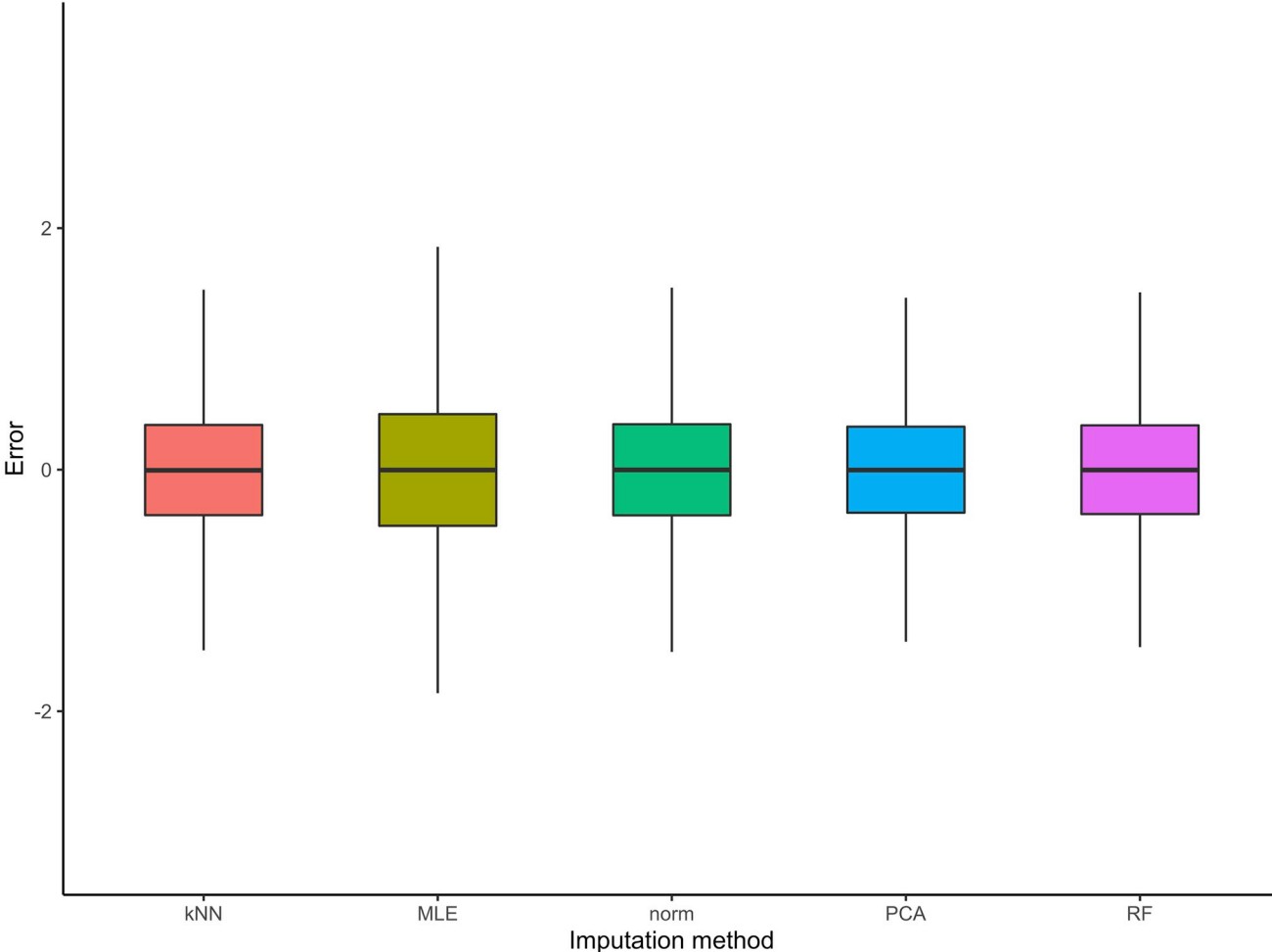

**Fig 5. Distribution of errors of the averaged imputed values for the five imputation methods considered on the second set of MAR simulations.**

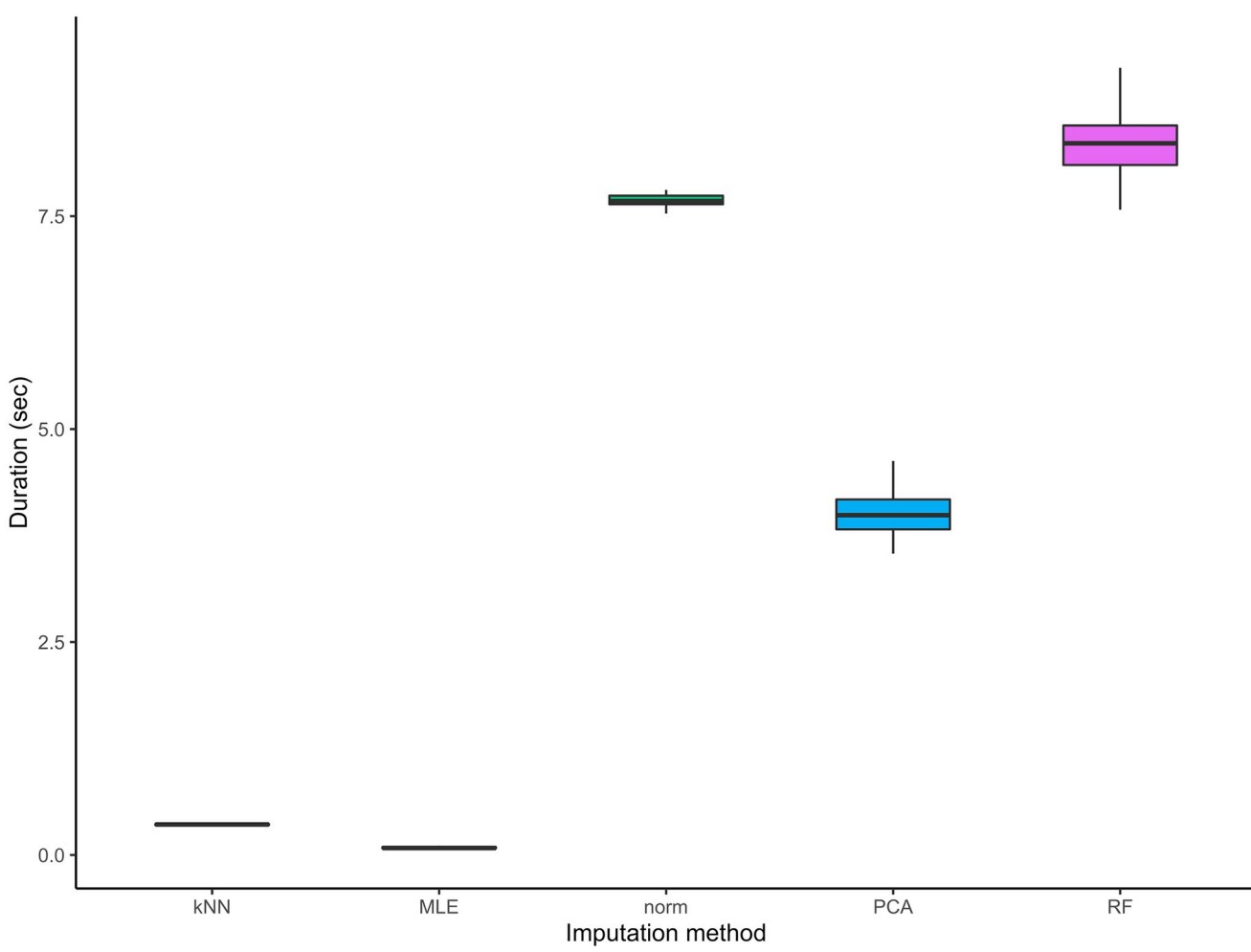

**Fig 6. Distributions of duration of the imputation process for the five imputation methods considered on the second set of MAR simulations.**

**Influence on testing results.**    The evaluation of performance for our `mi4p` methodology relies on the results produced by the testing procedure. For the MAR simulation designs, testing results were provided for all imputation methods considered. However, we could observe that no positives were produced for some datasets. As a summary, Table 2 describes under which conditions such pathological datasets arise in the second set of MAR simulations. The `mi4p` workflow dramatically underperforms at detecting positives when using the `norm` imputation method. The high number of pathological datasets can be explained by this method being a global one (*i.e.* applied to the full dataset), whereas other methods considered are local in that they are applied experimental condition-wise. Therefore, the `norm` method might lead to an increased between-imputation variability. Otherwise, no pathological cases occur while using the `mi4p` method on this particular set of simulated datasets. However, a few pathological datasets can be consistently observed when using the `DAPAR` workflow, regardless of the chosen imputation method. Overall, the `MLE` imputation offers a slight advantage over other methods.

**A glimpse of real datasets imputation.**    As a conclusion of this thorough analysis of synthetic data, let us draw some perspectives for the subsequent real datasets study. At this stage, `kNN` and `MLE` imputation methods might equivalently be considered. However, in quantitative

**Table 2. Number of pathological cases for each missing value proportion in the second set of MAR simulations.**

| Imputation method | Testing workflow | Missing value proportion | | | | | |
|---|---|---|---|---|---|---|---|
| | | 1% | 5% | 10% | 15% | 20% | 25% |
| kNN | DAPAR | 0 | 0 | 2 | 2 | 2 | 1 |
| | MI4P | 0 | 0 | 0 | 0 | 0 | 0 |
| MLE | DAPAR | 0 | 0 | 2 | 1 | 1 | 0 |
| | MI4P | 0 | 0 | 0 | 0 | 0 | 0 |
| norm | DAPAR | 0 | 0 | 2 | 2 | 1 | 0 |
| | MI4P | 0 | 0 | 0 | 7 | 26 | 57 |
| PCA | DAPAR | 0 | 0 | 2 | 2 | 3 | 0 |
| | MI4P | 0 | 0 | 0 | 0 | 0 | 0 |
| RF | DAPAR | 0 | 0 | 3 | 2 | 3 | 0 |
| | MI4P | 0 | 0 | 0 | 0 | 0 | 0 |

proteomics datasets, rows sometimes present more than 50% missing values. When this threshold is exceeded, current kNN method implementations only use mean imputation for these rows. However, mean imputation results in identical imputed values and no between-imputation variability arises, preventing from taking advantage of our mi4p methodology.

In contrast, the MLE imputation method still provides reliable imputations for a reduced computational cost in all situations. Moreover, the MLE method offers a more principled and interpretable approach compared to alternatives, which also motivated our choice to retain this method for further analysis of both MNAR + MCAR simulated datasets and real datasets.

**Experiments.** The distributions of the differences in sensitivity, specificity, precision, *F*-score and Matthews correlation coefficient between mi4p and DAPAR for all missing values proportion were summarised on the boxplots on Fig 7. Detailed results can be additionally found in S2 Table for MLE imputation and in S3, S4, S5 and S6 Tables for kNN, norm, PCA and RF imputations respectively. Both methods showed equivalent performance for a small proportion of missing values (1%), where the imputation process induces little variability. However, above 5% missing values, precision, *F*-Score and Matthews correlation coefficient were increasingly improved with the mi4p workflow compared to the DAPAR one. Moreover, sensitivity remained at 100% while specificity slightly improved, regardless of the missing value proportion. Note that the distributions of intensity values within each experimental condition for differentially expressed analytes are separate for the first set of MAR simulations. Indeed, intensity values for those analytes were drawn from a $\mathcal{N}(100, 1)$ distribution for the first condition and from a $\mathcal{N}(200, 1)$ distribution for the second one.

Compared to the first one, the second and the third sets of MAR simulations illustrate a case where the distributions of intensity values within each experimental condition for differentially expressed analytes are closer. Indeed, intensity values for these analytes were approximately drawn from a $\mathcal{N}(1.5, 0.5)$ distribution for the first condition and a $\mathcal{N}(3, 0.5)$ distribution for the second one. Fig 8 summarises the evolution of the distribution of differences in sensitivity, specificity, precision, *F*-score and Matthews correlation coefficient between mi4p and DAPAR depending on the proportion of missing values in the second set of MAR simulations. Detailed results can be additionally found in S7 Table for MLE imputation and in S8, S9, S10 and S11 Tables for kNN, norm, PCA and RF imputations respectively. A trade-off between sensitivity and specificity was observed for all proportions of missing values. Indeed, a slight loss in specificity (yet remaining above 99%) provided a greater gain in terms of sensitivity. However, precision performance remained equivalent in both methods.

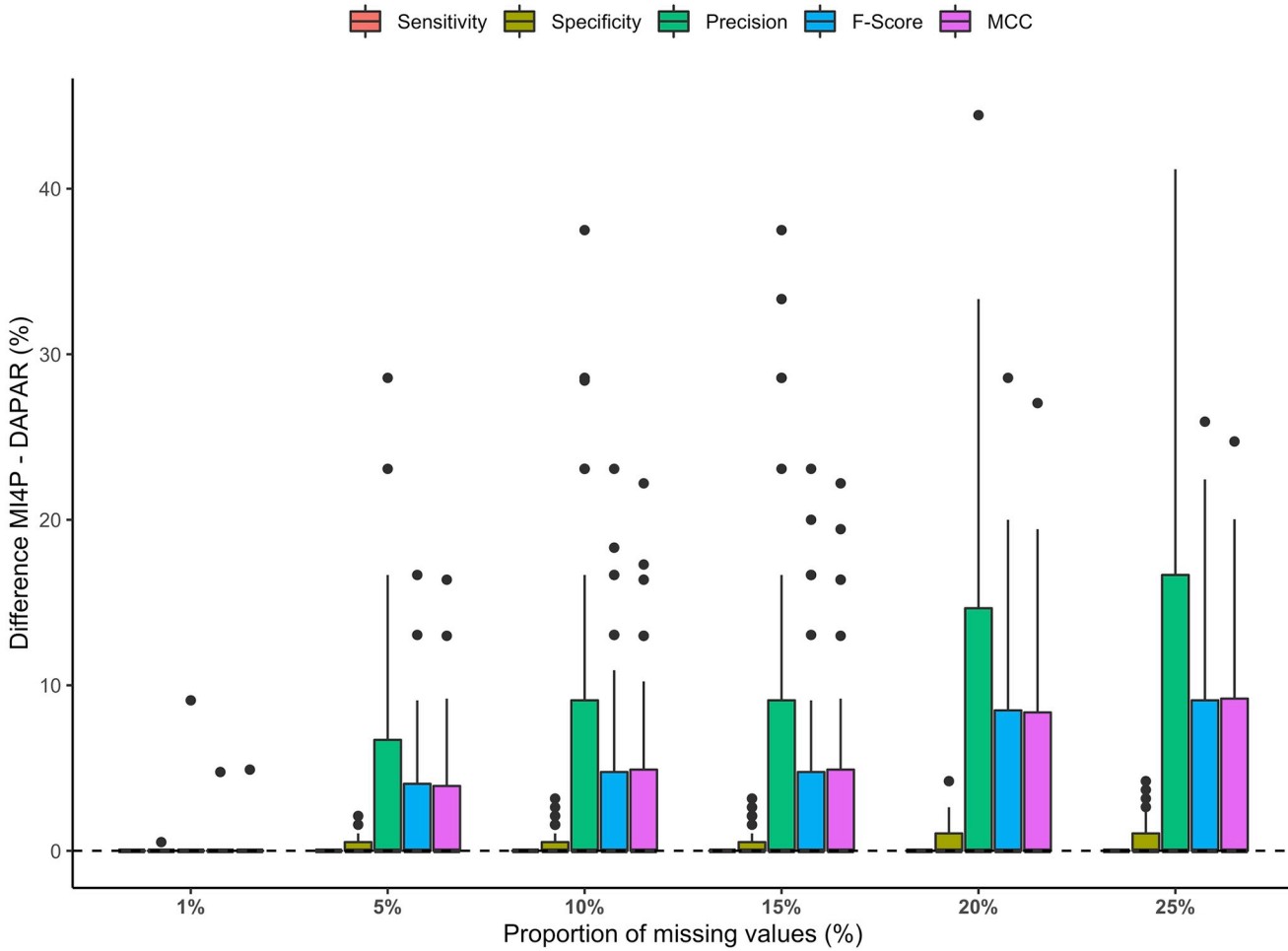

**Fig 7. Distributions of differences in sensitivity, specificity, precision, *F*-score and Matthews correlation coefficient for the first MAR set of simulations.** Missing values were imputed using the maximum likelihood estimation method.

Furthermore, the mean of *F*-scores and Matthews correlation coefficients across the 100 datasets were also increased with the `mi4p` workflow compared to the `DAPAR` one, suggesting a global improvement of the testing procedure's accuracy.

The third set of MAR simulations extended the second one from fixed to random effects. The difference in performance indicators represented in Fig 9 remained equivalent to the one observed in the previous set of simulations. However, the detailed results described in S12 Table suggested that both `mi4p` and `DAPAR` methods underperformed on data simulated based on random effects simulated data compared to the fixed effect simulation design. Detailed results can be additionally found in S12, S14, S15 and S16 Tables for `kNN`, `norm`, `PCA` and `RF` imputations respectively. Furthermore, the linear model on which both methods rely was not designed to account for random effects and thus struggles to capture such a source of variability. Therefore, an overall underperformance of both `mi4p` and `DAPAR` methods could be noticed in the third set of MAR simulations (S12 Table) compared to the second one (S7 Table).

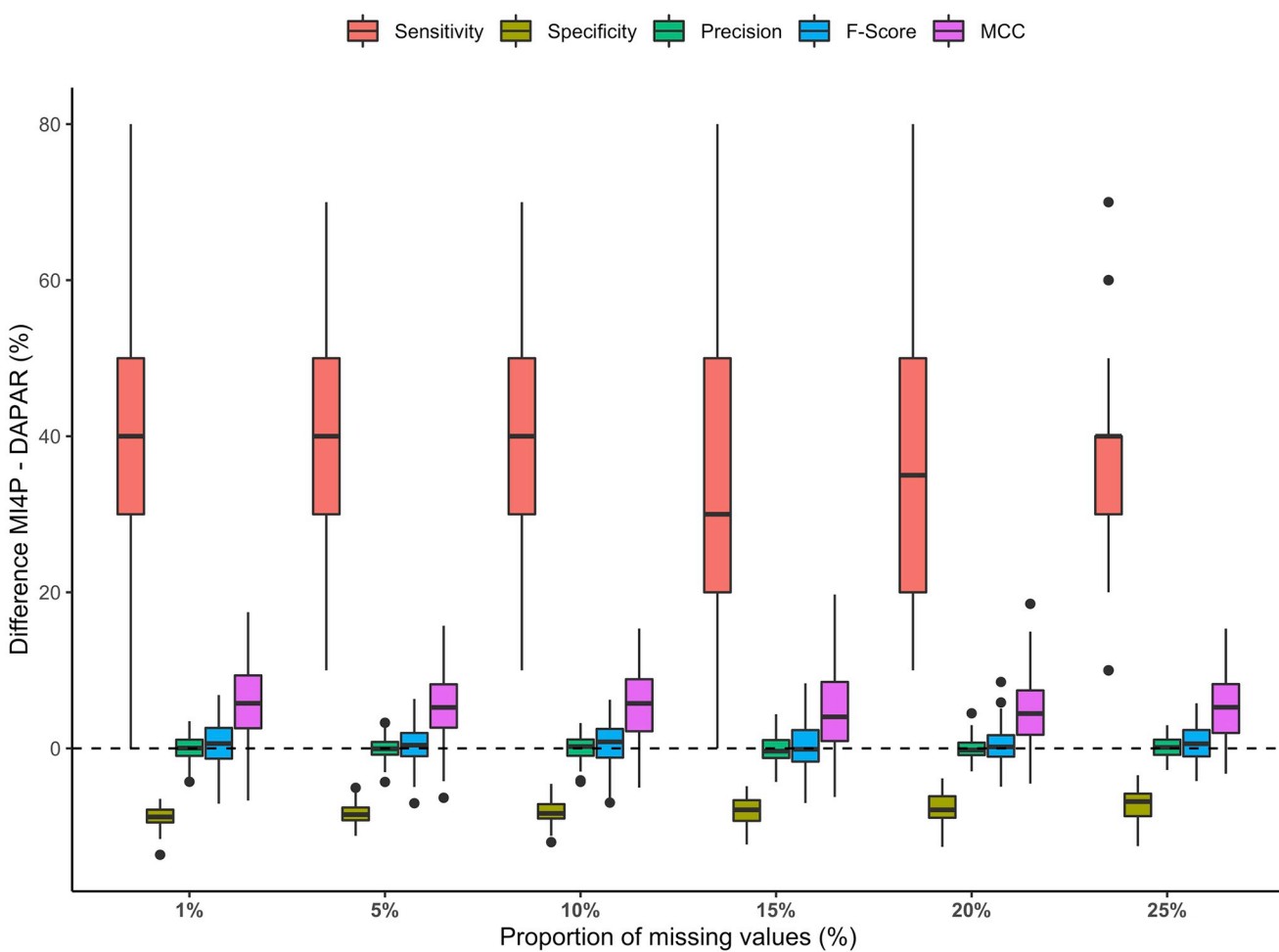

**Fig 8. Distributions of differences in sensitivity, specificity, precision, *F*-score and Matthews correlation coefficient for the second MAR set of simulations.** Missing values were imputed using the maximum likelihood estimation method.

## Simulated datasets under Missing completely at random and not at random assumption

**Simulation designs.**    The previous results were provided using only missing at random data. This section extends the simulation study to a mixture of missing completely at random (MCAR) and missing not at random (MNAR) data. The data were simulated following an experimental design implemented in the imp4p R package through the sim.data function [14, 15].

The first set of simulations was based on the following experimental design. Two experimental conditions with ten biological samples each were considered, for which the log-intensities of 1000 analytes were simulated. Among them, 200 were set to be differentially expressed. Hence, the 200 differentially expressed analytes have log-intensities drawn from a Gaussian distribution with a mean of 12.5 in the first condition and 25 in the second one. The remaining simulated log-intensities of non differentially expressed analytes are drawn for both conditions from a Gaussian distribution with a mean of 12.5. The standard deviation in each condition for all analytes is set to 2. Other parameters to be passed as arguments in the sim.data function were set to default values.

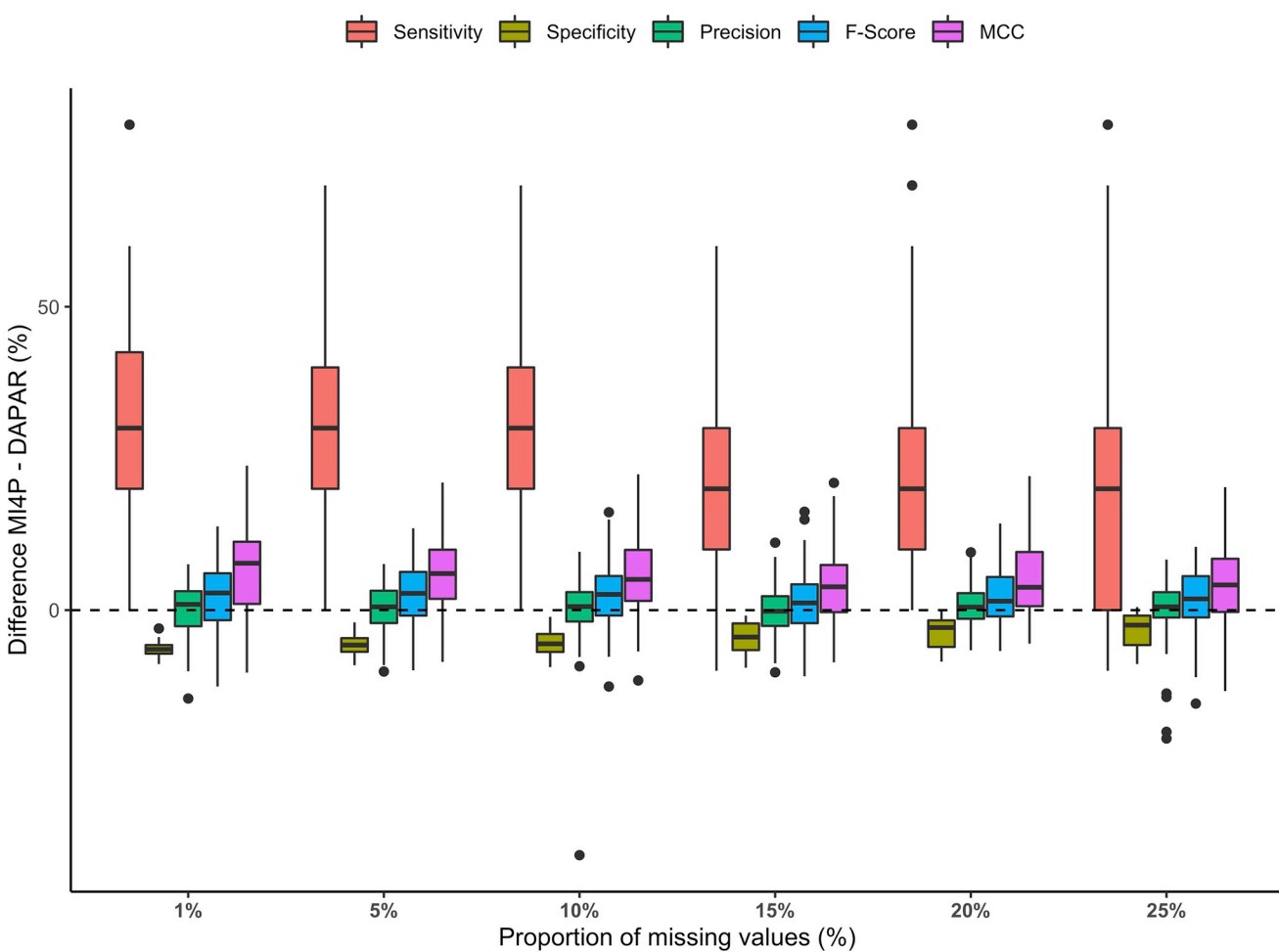

**Fig 9. Distributions of differences in sensitivity, specificity, precision, *F*-score and Matthews correlation coefficient for the third MAR set of simulations.** Missing values were imputed using the maximum likelihood estimation method.

The second set of simulations considered extends the first one by increasing the number of simulated analytes to 10,000, among which 500 are differentially expressed. Note that in this design, the proportion of differentially expressed analytes is decreased from 20% to 5%. For both simulation studies, six datasets were built with 1%, 5%, 10%, 20% and 25% missing values.

**Experiments.**   The distributions of the difference of the previously described indicators of performance between the `mi4p` and the `DAPAR` workflows for the first set of simulations were shown in Fig 10. A trade-off between sensitivity and specificity could be observed: sensitivity was increased by 15% on average while specificity was decreased by 15% on average for the `mi4p` workflow compared to the `DAPAR` one. Furthermore, performance in terms of precision was equivalent for both methods. As far as global performances are concerned, the *F*-Score was slightly increased by an average of 2%, and the MCC was quite stable, with a slight decrease observed for the data with the highest missing values proportion.

Fig 11 depicts the distributions of the difference of the previously described indicators of performance between the `mi4p` and the `DAPAR` workflows for the second set of simulations. The dispersions of the distributions are globally reduced, but the same trends as in the first set of simulations can be observed. Detailed results for both sets of simulations can be found in

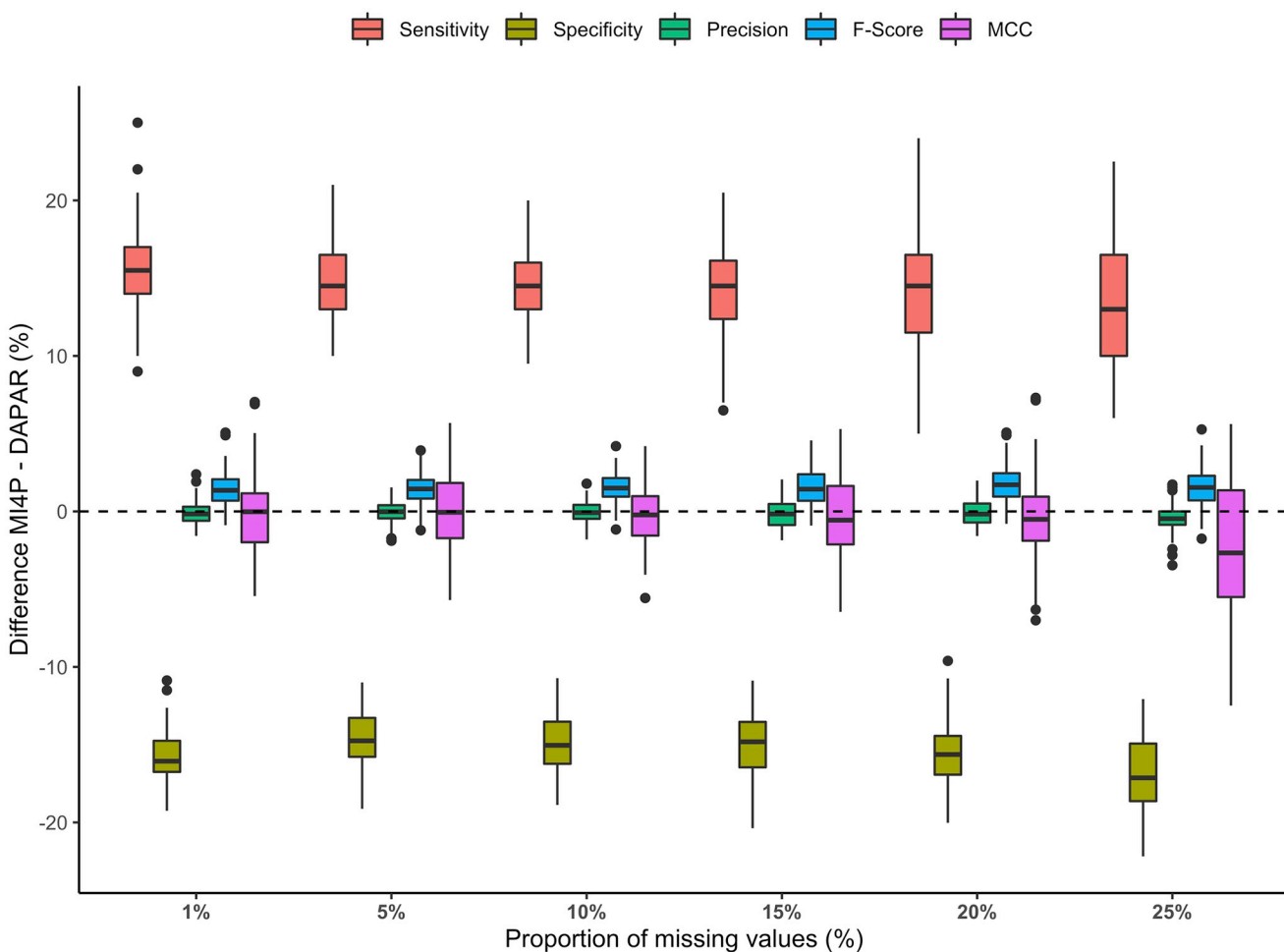

**Fig 10. Distributions of differences in sensitivity, specificity, precision, *F*-score and Matthews correlation coefficient for the first MCAR + MNAR set of simulations.** Missing values were imputed using the maximum likelihood estimation method.

S17 and S18 Tables. Overall performance in terms of sensitivity, specificity, and precision is quite low for both `mi4p` and `DAPAR` methods, mainly due to a large number of false positives. In particular, precision performance drops when the number of analytes considered is increased. Moreover, the poor performance in terms of MCC suggests that both methods behave almost as random guess classifiers. Hence, the relevance of the chosen imputation method should be questioned in this framework.

Simulation studies showed more false positives in datasets with MNAR and MCAR values than with MAR values. While the considered datasets were simulated differently, this observation requires further investigation, particularly on the imputation method used. Recent works suggested that a combination of MCAR-devoted and MNAR-devoted imputation algorithms perform most accurately and reproducibly on bottom-up proteomics data regardless of the missing value type (except for high MNAR proportions) [14, 33].

## Real quantitative proteomics datasets

**Controlled datasets generation.** **Complex total cell lysates spiked UPS1 standard protein mixtures.** We consider a first real dataset based on the following experiment. Six

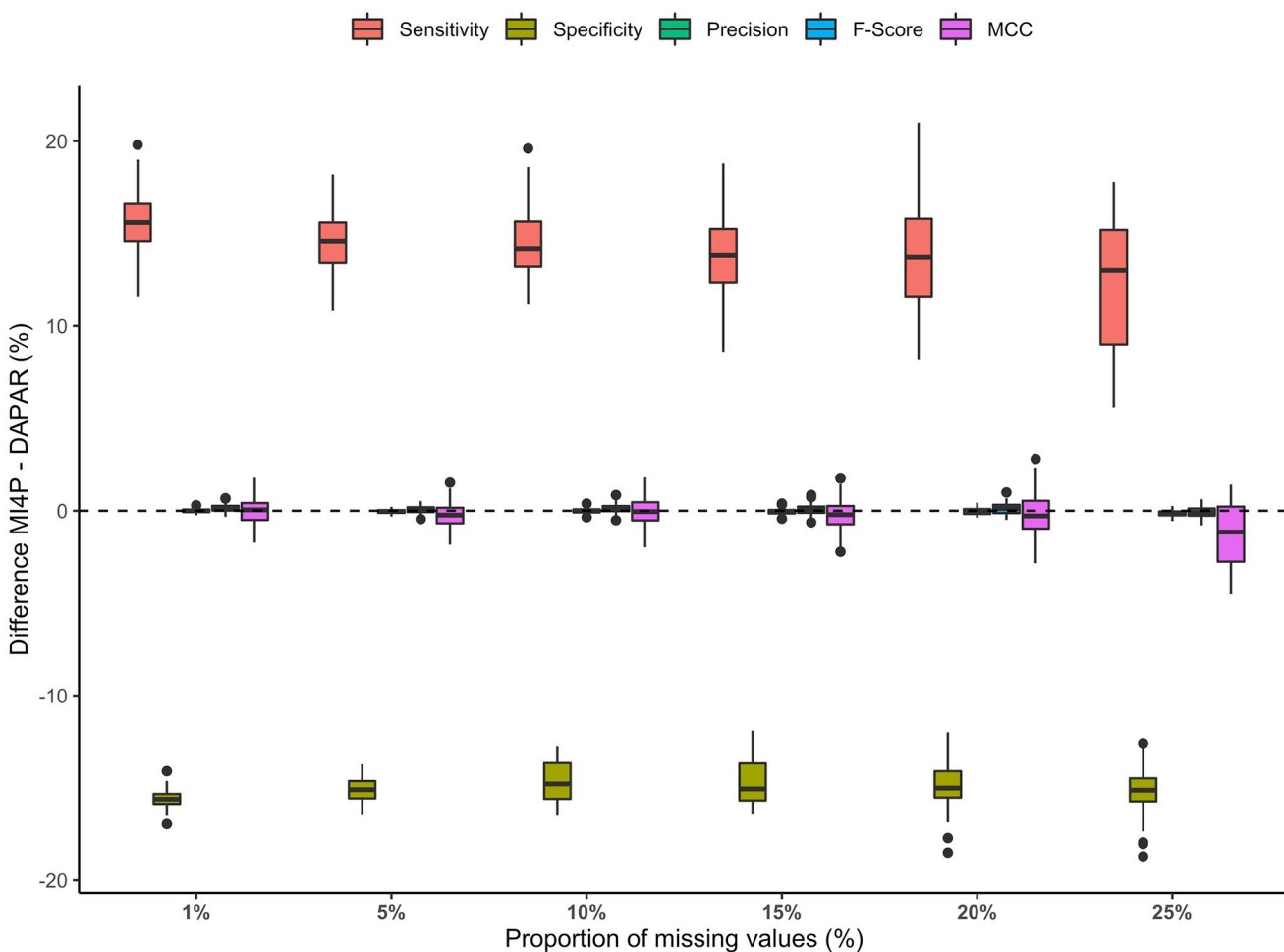

**Fig 11. Distributions of differences in sensitivity, specificity, precision, *F*-score and Matthews correlation coefficient for the second MCAR + MNAR set of simulations.** Missing values were imputed using the maximum likelihood estimation method.

peptide mixtures, composed of a constant yeast (*Saccharomyces cerevisiae*) background, into which increasing amounts of UPS1 standard proteins (48 recombinant human proteins, Merck) were spiked at 0.5, 1, 2.5, 5, 10 and 25 fmol, respectively [34]. In a second well-calibrated dataset, yeast was replaced by a more complex total lysate of *Arabidopsis thaliana* in which UPS1 was spiked in 7 different amounts, namely 0.05, 0.25, 0.5, 1.25, 2.5, 5 and 10 fmol. For each mixture, technical triplicates were constituted. The *Saccharomyces cerevisiae* dataset was acquired on a nanoLC-MS/MS coupling composed of a nanoAcquity UPLC device (Waters) coupled to a Q-Exactive Plus mass spectrometer (Thermo Scientific, Bremen, Germany) [34]. The *Arabidopsis thaliana* dataset was acquired on a nanoLC-MS/MS coupling composed of nanoAcquity UPLC device (Waters) coupled to a Q-Exactive HF-X mass spectrometer (Thermo Scientific, Bremen, Germany) as described hereafter.

**Data preprocessing.** For the *Saccharomyces cerevisiae* and *Arabidopsis thaliana* datasets, Maxquant software was used to identify peptides and derive extracted ion chromatograms. Peaks were assigned with the Andromeda search engine with full trypsin specificity. The database used for the searches was concatenated in house with the *Saccharomyces cerevisiae* entries extracted from the UniProtKB-SwissProt database (16 April 2015, 7806 entries) or the *Arabidopsis thaliana* entries (09 April 2019, 15 818 entries) and those of the UPS1 proteins (48

entries). The minimum peptide length required was seven amino acids and a maximum of one missed cleavage was allowed. Default mass tolerances parameters were used. The maximum false discovery rate was 1% at peptide and protein levels with the use of a decoy strategy. For the *Arabidopsis thaliana* + UPS1 experiment, data were extracted both with and without Match Between Runs and 2 pre-filtering criteria were applied prior to statistical analysis: only peptides with at least 1 out of 3 quantified values in each condition on one hand and 2 out of 3 on the other hand were kept. Thus, 4 datasets derived from the *Arabidopsis thaliana* + UPS1 were considered. For the *Saccharomyces cerevisiae* + UPS1 experiment, the same filtering criteria were applied, but only on data extracted with Match Between Runs, leading to 2 datasets considered. An additional normalisation step was performed on each dataset considered. Normalising peptides' or proteins' intensities aims at reducing batch effects, sample-level variations and therefore better comparing intensities across studied biological samples [35]. In this work, we chose to perform quantile normalisation [36], using the `normalize.quantiles` function from the `preprocessCore` R package [37].

**Supplemental methods for *Arabidopsis thaliana* dataset.**   Peptide separation was performed on an ACQUITY UPLC BEH130 C18 column (250 mm × 75 $\mu$m with 1.7 $\mu$m diameter particles) and a Symmetry C18 precolumn (20 mm ×180 $\mu$m with 5 $\mu$m diameter particles; Waters). The solvent system consisted of 0.1% FA in water (solvent A) and 0.1% FA in ACN (solvent B). The samples were loaded into the enrichment column over 3 min at 5 $\mu$L/min with 99% of solvent A and 1% of solvent B. The peptides were eluted at 400 nL/min with the following gradient of solvent B: from 3 to 20% over 63 min, 20 to 40% over 19 min, and 40 to 90% over 1 min. The MS capillary voltage was set to 2kV at 250˚C. The system was operated in a data-dependent acquisition mode with automatic switching between MS (mass range 375–1500 m/z with R = 120 000, automatic gain control fixed at $3 \times 106$ ions, and a maximum injection time set at 60 ms) and MS/MS (mass range 200–2000 m/z with R = 15 000, automatic gain control fixed at $1 \times 105$, and the maximal injection time set to 60 ms) modes. The twenty most abundant peptides were selected on each MS spectrum for further isolation and higher energy collision dissociation fragmentation, excluding unassigned and monocharged ions. The dynamic exclusion time was set to 40s.

**Experiments.**   The trade-off suggested by the simulation study was confirmed by the results obtained on the real datasets. In the *Saccharomyces cerevisiae* + UPS1 experiment, a decrease of 70% in the number of false positives was observed, improving the specificity and precision (see S25 Table), at the cost of the number of true positives (Table 3), thus decreasing the sensitivity.

The same trend is observed in the *Arabidopsis thaliana* + UPS1 experiment; the number of false positives is decreased by 50% (see Table 4 and S19 Table), thus improving specificity and precision at the cost of sensitivity. The loss in sensitivity is larger in the highest points of the range in both experiments. The structure of the calibrated datasets used here can explain these observations. Indeed, the quantitative dataset considered takes into account all samples from all conditions, while the testing procedure focuses on one-vs-one comparisons. Two issues can be raised:

**Table 3. Performance of the `mi4p` methodology expressed in percentage with respect to `DAPAR` workflow, on *Saccharomyces cerevisiae* + UPS1 experiment, with Match Between Runs and at least 1 out of 3 quantified values in each condition.** Missing values (6%) were imputed using the maximum likelihood estimation method.

| Condition vs. 25fmol | True positives | False positives | Sensitivity | Specificity | F-Score |
|:---:|:---:|:---:|:---:|:---:|:---:|
| 0.5fmol | -2.7% | -67.2% | -2.7% | +1.6% | +53.6% |
| 1fmol | -1.6% | -71.1% | -0.5% | +0.9% | +37.8% |
| 2.5fmol | -3.2% | -75.8% | -3.3% | +0.7% | +26.9% |
| 5fmol | -14.3% | -78.7% | -14.3% | +0.5% | +11.4% |
| 10fmol | -41.9% | -75.2% | -41.9% | +0.5% | -14.4% |

**Table 4. Performance of the `mi4p` methodology expressed in percentage with respect to `DAPAR` workflow, on *Arabidopsis thaliana* + UPS1 experiment, with at least 1 out of 3 quantified values in each condition.** Missing values (6%) were imputed using the maximum likelihood estimation method.

| Condition vs. 10fmol | True positives | False positives | Sensitivity | Specificity | F-Score |
|:---:|:---:|:---:|:---:|:---:|:---:|
| 0.05fmol | -2.3% | -43% | -2.3% | +15% | +62.7% |
| 0.25fmol | -1.5% | -43% | -1.4% | +13.9% | +65.3% |
| 0.5fmol | -1.5% | -50.6% | -1.4% | +10.8% | +81.4% |
| 1.25fmol | -2.3% | -62.6% | -2.3% | +10.9% | +119.8% |
| 2.5fmol | -25.6% | -69.3% | -25.5% | +2.4% | +45.9% |
| 5fmol | -30.3% | -65.2% | -30.4% | +5.5% | +56.1% |

- The data preprocessing step can lead to more data filtering than necessary. For instance, we chose to use the filtering criterion such that rows with at least one quantified value in each condition were kept. The more conditions are considered, the more stringent the rule is, possibly leading to a poorer dataset (with fewer observations) for the conditions of interest.

- The imputation process is done on the whole dataset, as well as the estimation step. Then, while projecting the variance-covariance matrix, the estimated variance (later used in the test statistic) is the same for all comparisons. Thus, if one is interested in comparing conditions with fewer missing values, the variance estimator will be penalised by the presence of conditions with more missing values in the initial dataset.

This phenomenon was illustrated in S20 Table, where solely the two highest points of the range have been compared, only using the quantitative data from those two conditions. Hence, more peptides have been taken into account for the statistical analysis. This strategy led to overall better scores for precision, *F*-score and Matthews correlation coefficient compared to the previous framework.

As far as data extracted without the Match Between Runs algorithm are concerned, the results were equivalent in both methods considered in the *Arabidopsis thaliana* + UPS1 experiment (as illustrated in S22 and S23 Tables). Furthermore, the same observations could be drawn from datasets filtered with the criterion of a minimum of 2 out of 3 observed values in each group for the *Arabidopsis thaliana* + UPS1 experiment (S21 and S23 Tables) as well as for the *Saccharomyces cerevisiae* + UPS1 experiment (S26 Table). These observations translated a loss of global information in the dataset, as filtering criteria led to fewer peptides considered with fewer missing values per peptide.

The `mi4p` methodology also provided better results at the protein-level (after aggregation) in terms of specificity, precision, *F*-score and Matthews correlation coefficient, with a minor loss in sensitivity (S27 Table). In particular, a decrease of 63.2% to 80% in the number of false positives was observed with a lower loss on the number of true positives and on sensitivity (up to 2.6%) for the *Saccharomyces cerevisiae* + UPS1 experiment, as illustrated in Table 5. As far as the *Arabidopsis thaliana* + UPS1 experiment is concerned, the same trend was observed (S24 Table). Indeed, the number of false positives was decreased by 31% to 66.8%, with a maximum loss in the number of true positives of 9.8%, as illustrated in Table 6.

## Conclusion

In this work, we presented a rigorous multiple imputation method as a key step of a workflow by combining the imputed datasets using Rubin's rules. We thus obtained for each analyte, on the one hand, a combined estimator of the vector of interest parameters, and on the other

**Table 5. Performance of the `mi4p` methodology (with the aggregation step) expressed in percentage with respect to `DAPAR` workflow, on *Saccharomyces cerevisiae* + UPS1 experiment, with at least 1 out of 3 quantified values in each condition.** Missing values were imputed using the Maximum Likelihood Estimation method.

| Condition vs. 25fmol | True positives | False positives | Sensitivity | Specificity | F-Score |
|---|---|---|---|---|---|
| 0.5fmol | 0% | -73.3% | 0% | +2.9% | +61.1% |
| 1fmol | -2.4% | -80% | -2.4% | +2.3% | +51.4% |
| 2.5fmol | 0% | -70.4% | 0% | +0.8% | +20.9% |
| 5fmol | -2.4% | -63.2% | -2.4% | +0.5% | +11.6% |
| 10fmol | -2.6% | -69.6% | -2.6% | +0.7% | +16.5% |

**Table 6. Performance of the `mi4p` methodology (with the aggregation step) expressed in percentage with respect to `DAPAR` workflow, on *Arabidopsis thaliana* + UPS1 experiment, with at least 1 out of 3 quantified values in each condition.** Missing values were imputed using the Maximum Likelihood Estimation method.

| Condition vs. 10fmol | True positives | False positives | Sensitivity | Specificity | F-Score |
|---|---|---|---|---|---|
| 0.05fmol | 0% | -27.6% | 0% | +18.3% | +34.2% |
| 0.25fmol | 0% | -25.7% | 0% | +18.1% | +31% |
| 0.5fmol | 0% | -31% | 0% | +15.2% | +39.5% |
| 1.25fmol | 0% | -65.3% | 0% | +12.1 | +119.2% |
| 2.5fmol | -2.4% | -66.8% | -2.4% | +5.8% | +88.3% |
| 5fmol | -9.8% | -57.3% | -9.8% | +12.9% | +78.9% |

hand, an estimator of its corresponding variance-covariance matrix. Hence, both within- and between-imputation variabilities are accounted for. The variance-covariance matrix was projected to get a univariate variance parameter for each analyte. We then considered this variability downstream of the statistical analysis by including it in the well-known moderated $t$-test statistic. In addition, we provided insights on the comparison of imputation methods. Our methodology was implemented in a publicly available R package named `mi4p`. Its performance was compared on both simulated and real datasets to the `DAPAR` state-of-the-art methodology, using confusion matrix-based indicators. The results showed a trade-off between those indicators. In real datasets, the methodology reduces the number of false positives in exchange for a minor reduction of the number of true positives. The results are similar among all imputation methods considered, especially when the proportion of missing values is small. Our methodology with an additional aggregation step provides better results with a minor loss in sensitivity and can be of interest for proteomicists who will benefit from results at the protein level while using peptide-level quantification data.

## Supporting information

**S1 Table. State of the art on imputation in quantitative proteomics.** This table gives an overview of the recent literature on imputation methods in quantitative proteomics.
(PDF)

**S2 Table. Performance evaluation on the first set of MAR simulations imputed using maximum likelihood estimation.** Results are provided as mean ± standard deviation over the 100 simulated datasets for each indicator of performance.
(PDF)

**S3 Table. Performance evaluation on the first set of MAR simulations imputed using *k*-nearest neighbours.** Results are provided as mean ± standard deviation over the 100 simulated

datasets for each indicator of performance.
(PDF)

**S4 Table. Performance evaluation on the first set of MAR simulations imputed using Bayesian linear regression.** Results are provided as mean ± standard deviation over the 100 simulated datasets for each indicator of performance.
(PDF)

**S5 Table. Performance evaluation on the first set of MAR simulations imputed using principal component analysis.** Results are provided as mean ± standard deviation over the 100 simulated datasets for each indicator of performance.
(PDF)

**S6 Table. Performance evaluation on the first set of MAR simulations imputed using random forests.** Results are provided as mean ± standard deviation over the 100 simulated datasets for each indicator of performance.
(PDF)

**S7 Table. Performance evaluation on the second set of MAR simulations imputed using maximum likelihood estimation.** Results are provided as mean ± standard deviation over the 100 simulated datasets for each indicator of performance.
(PDF)

**S8 Table. Performance evaluation on the second set of MAR simulations imputed using $k$-nearest neighbours.** Results are provided as mean ± standard deviation over the 100 simulated datasets for each indicator of performance.
(PDF)

**S9 Table. Performance evaluation on the second set of MAR simulations imputed using Bayesian linear regression.** Results are provided as mean ± standard deviation over the 100 simulated datasets for each indicator of performance.
(PDF)

**S10 Table. Performance evaluation on the second set of MAR simulations imputed using principal component analysis.** Results are provided as mean ± standard deviation over the 100 simulated datasets for each indicator of performance.
(PDF)

**S11 Table. Performance evaluation on the second set of MAR simulations imputed using random forests.** Results are provided as mean ± standard deviation over the 100 simulated datasets for each indicator of performance.
(PDF)

**S12 Table. Performance evaluation on the third set of MAR simulations imputed using maximum likelihood estimation.** Results are provided as mean ± standard deviation over the 100 simulated datasets for each indicator of performance.
(PDF)

**S13 Table. Performance evaluation on the third set of MAR simulations imputed using $k$-nearest neighbours.** Results are provided as mean ± standard deviation over the 100 simulated datasets for each indicator of performance.
(PDF)

**S14 Table. Performance evaluation on the third set of MAR simulations imputed using Bayesian linear regression.** Results are provided as mean ± standard deviation over the 100 simulated datasets for each indicator of performance.
(PDF)

**S15 Table. Performance evaluation on the third set of MAR simulations imputed using principal component analysis.** Results are provided as mean ± standard deviation over the 100 simulated datasets for each indicator of performance.
(PDF)

**S16 Table. Performance evaluation on the third set of MAR simulations imputed using random forests.** Results are provided as mean ± standard deviation over the 100 simulated datasets for each indicator of performance.
(PDF)

**S17 Table. Performance evaluation on the first set of MCAR + MNAR simulations imputed using maximum likelihood estimation.** Results are provided as mean ± standard deviation over the 100 simulated datasets for each indicator of performance.
(PDF)

**S18 Table. Performance evaluation on the second set of MCAR + MNAR simulations imputed using maximum likelihood estimation.** Results are provided as mean ± standard deviation over the 100 simulated datasets for each indicator of performance.
(PDF)

**S19 Table. Performance evaluation on the *Arabidopsis thaliana* + UPS1 dataset, filtered with at least 1 quantified value in each condition.** Missing values were imputed using the maximum likelihood estimation method.
(PDF)

**S20 Table. Performance evaluation on the *Arabidopsis thaliana* + UPS1 dataset, filtered with at least 1 quantified value in each condition and focusing only on the comparison 5fmol vs. 10fmol.** Missing values were imputed using the maximum likelihood estimation method.
(PDF)

**S21 Table. Performance evaluation on the *Arabidopsis thaliana* + UPS1 dataset, filtered with at least 2 quantified values in each condition.** Missing values were imputed using the maximum likelihood estimation method.
(PDF)

**S22 Table. Performance evaluation on the *Arabidopsis thaliana* + UPS1 dataset, extracted without Match Between Runs and filtered with at least 1 quantified value in each condition.** Missing values were imputed using the maximum likelihood estimation method.
(PDF)

**S23 Table. Performance evaluation on the *Arabidopsis thaliana* + UPS1 dataset, extracted without Match Between Runs and filtered with at least 2 quantified value in each condition.** Missing values were imputed using the maximum likelihood estimation method.
(PDF)

**S24 Table. Performance evaluation on the *Arabidopsis thaliana* + UPS1 dataset at the protein-level, filtered with at least 1 quantified values in each condition.** Missing values were

imputed using the maximum likelihood estimation method.
(PDF)

**S25 Table. Performance evaluation on the *Saccharomyces cerevisiae* + UPS1 dataset, filtered with at least 1 quantified value in each condition.** Missing values were imputed using the maximum likelihood estimation method.
(PDF)

**S26 Table. Performance evaluation on the *Saccharomyces cerevisiae* + UPS1 dataset, filtered with at least 2 quantified values in each condition.** Missing values were imputed using the maximum likelihood estimation method.
(PDF)

**S27 Table. Performance evaluation on the *Saccharomyces cerevisiae* + UPS1 dataset, at the protein-level and filtered with at least 1 quantified values in each condition.** Missing values were imputed using the maximum likelihood estimation method.
(PDF)

## Acknowledgments

The authors wish to thank Leslie Muller and Nicolas Pythoud for providing the real proteomics datasets used in this work, as well as Thomas Burger and Quentin Giai-Gianetto for their help on the DAPAR and imp4p R packages.

## Author Contributions

**Conceptualization:** Marie Chion, Christine Carapito, Frédéric Bertrand.

**Data curation:** Marie Chion.

**Formal analysis:** Marie Chion.

**Funding acquisition:** Christine Carapito, Frédéric Bertrand.

**Investigation:** Marie Chion.

**Methodology:** Marie Chion, Frédéric Bertrand.

**Project administration:** Christine Carapito, Frédéric Bertrand.

**Resources:** Marie Chion.

**Software:** Marie Chion, Frédéric Bertrand.

**Supervision:** Christine Carapito, Frédéric Bertrand.

**Validation:** Marie Chion, Christine Carapito.

**Visualization:** Marie Chion.

**Writing – original draft:** Marie Chion, Christine Carapito, Frédéric Bertrand.

**Writing – review & editing:** Marie Chion, Christine Carapito, Frédéric Bertrand.

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
