## [Decision Letter · Decision Letter 0]

11 Apr 2022

Dear Dr. Chion,

Thank you very much for submitting your manuscript "Accounting for multiple imputation-induced variability for differential analysis in mass spectrometry-based label-free quantitative proteomics" for consideration at PLOS Computational Biology.

As with all papers reviewed by the journal, your manuscript was reviewed by members of the editorial board and by several independent reviewers. In light of the reviews (below this email), we would like to invite the resubmission of a significantly-revised version that takes into account the reviewers' comments.

As recommended by the reviewers, it would be useful to contextualize mi4p better within the state of the art for proteomics imputation. Additionally, the reviewers raise important concerns related to the simulated datasets that should be addressed. Finally, several changes are recommended to improve the usability of the mi4p software, also for non-statistical users, and provide full code details associated with the manuscript.

We cannot make any decision about publication until we have seen the revised manuscript and your response to the reviewers' comments. Your revised manuscript is also likely to be sent to reviewers for further evaluation.

Sincerely,

Wout Bittremieux

Guest Editor

PLOS Computational Biology

William Noble

Deputy Editor

PLOS Computational Biology

Reviewer's Responses to Questions

**Comments to the Authors:**

Reviewer #1: I have uploaded my review as a pdf file.

Reviewer #2: The authors developed a multiple imputation strategy to get a better performance than the single imputation strategy. In the results section, they simulated datasets with missing at random and missing not at random assumptions to test the performance of their method. The results show that their method performed better than DAPAR. They also used real dataset to confirm the conclusion. Overall, this manuscript was well written and organized. The performance of the tool seems promising.

Reviewer #3: Comments attached

**Have the authors made all data and (if applicable) computational code underlying the findings in their manuscript fully available?**

Reviewer #1: Yes

Reviewer #2: Yes

Reviewer #3: **No: **No code for data simulations or data analysis were provided

PLOS authors have the option to publish the peer review history of their article (what does this mean?). If published, this will include your full peer review and any attached files.

Reviewer #1: **Yes: **Ludger Goeminne

Reviewer #2: No

Reviewer #3: No
---

## [Decision Letter · Decision Letter 1]

6 Jul 2022

Dear Dr. Chion,

Thank you very much for submitting your manuscript "Accounting for multiple imputation-induced variability for differential analysis in mass spectrometry-based label-free quantitative proteomics" for consideration at PLOS Computational Biology.

The reviewers appreciated your thorough efforts in submitting your revised manuscript. As you will see, there are still a few minor recommendations for clarifications. As soon as these have been provided, we will accept this manuscript for publication.

Sincerely,

Wout Bittremieux

Guest Editor

PLOS Computational Biology

William Noble

Deputy Editor

PLOS Computational Biology

[LINK]

Reviewer's Responses to Questions

**Comments to the Authors:**

Reviewer #1: Dear editor

I would like to congratulate the authors on their thorough revision. My comments have been accurately addressed.

I only have some minor comments on the newly inserted paragraph, which I think could be addressed relatively smoothly.

Comment 1:

I am not sure what is meant by "pathological cases" in table 2 and the text preceding it.

Do the authors mean the number of "false positives"? If so, they should use this terminology.

If this interpretation is correct, I wonder why the authors picked this metric and not e.g. sensitivity, specificity or F-score for example? (this could be addressed by adding a sentence explaining why the authors compare these methods in terms of the numbers of false positives)

Comment 2:

Line 275: the authors state "The mi4p workflow dramatically underperforms at detecting positives when using the norm imputation method."

I assume the term "positives" refers to "true positives"? If this interpretation is correct, is there somewhere a table with the number of true positives to back up this claim? Alternatively, if this sentence refers to the high number of false positives, I would rephrase this sentence accordingly, e.g. by saying that the mi4p workflow detects a lot of false positives when using the norm imputation method.

Comment 3:

I would also suggest to explicitly add a field to table 2 to make it clear from the table itself that the given percentages are the amputation percentages.

Comment 4:

I further have some small grammar suggestions:

Line 105: "coefficient of the linear model pour peptide p" -> "coefficient of the linear model for peptide p"

Line 264: "a slightly increased variability than other methods" -> "a slightly increased variability compared to other methods"

Line 270: remove "are compared as well"

Line 271: "for imputing DQ times each simulated dataset" -> "for imputing each simulated dataset DQ times"

Line 272: remove "According to"

Reviewer #3: The revised manuscript is greatly improved, and the authors have satisfactorily addressed all of my concerns.

**Have the authors made all data and (if applicable) computational code underlying the findings in their manuscript fully available?**

Reviewer #1: Yes

Reviewer #3: Yes

PLOS authors have the option to publish the peer review history of their article (what does this mean?). If published, this will include your full peer review and any attached files.

Reviewer #1: **Yes: **Ludger Goeminne

Reviewer #3: No

Figure Files:

Data Requirements:

Reproducibility:

References:

---

## [Editor Report · Decision Letter 2]

21 Jul 2022

Dear Dr. Chion,

We are pleased to inform you that your manuscript 'Accounting for multiple imputation-induced variability for differential analysis in mass spectrometry-based label-free quantitative proteomics' has been provisionally accepted for publication in PLOS Computational Biology.

Best regards,

Wout Bittremieux

Guest Editor

PLOS Computational Biology

William Noble

Deputy Editor

PLOS Computational Biology

---

## [Editor Report · Acceptance letter]

25 Aug 2022

PCOMPBIOL-D-22-00385R2 

Accounting for multiple imputation-induced variability for differential analysis in mass spectrometry-based label-free quantitative proteomics

Dear Dr Chion,

I am pleased to inform you that your manuscript has been formally accepted for publication in PLOS Computational Biology. Your manuscript is now with our production department and you will be notified of the publication date in due course.

With kind regards,

Zsuzsanna Gémesi
